# Noise-Reduction and Sensitivity-Enhancement of a Sleeping Beauty-Based Tet-On System

**DOI:** 10.3390/genes13101679

**Published:** 2022-09-20

**Authors:** Sarah C. Saunderson, SM Ali Hosseini-Rad, Alexander D. McLellan

**Affiliations:** 1Department of Microbiology and Immunology, University of Otago, Dunedin 9016, New Zealand; 2Centre of Excellence in Immunology and Immune-Mediated Diseases, University of Chulalongkorn, Bangkok 10330, Thailand

**Keywords:** tetracycline inducible promoter, sleeping beauty transposon, gene regulation

## Abstract

Tetracycline-inducible systems are widely used control elements for mammalian gene expression. Despite multiple iterations to improve inducibility, their use is still compromised by basal promoter activity in the absence of tetracyclines. In a mammalian system, we previously showed that the introduction of the G72V mutation in the rtTA-M2 tetracycline activator lowers the basal level expression and increases the fold-induction of multiple genetic elements in a long chimeric antigen receptor construct. In this study, we confirmed that the G72V mutation was effective in minimising background expression in the absence of an inducer, resulting in an increase in fold-expression. Loss of responsiveness due to the G72V mutation was compensated through the incorporation of four sensitivity enhancing (SE) mutations, without compromising promoter tightness. However, SE mutations alone (without G72V) led to undesirable leakiness. Although cryptic splice site removal from rtTA did not alter the inducible control of the luciferase reporter gene in this simplified vector system, this is still recommended as a precaution in more complex multi-gene elements that contain rtTA. The optimized expression construct containing G72V and SE mutations currently provides the best improvement of fold-induction mediated by the rtTA-M2 activator in a mammalian system.

## 1. Introduction

Tetracycline-activated gene expression allows for the control of a variety of coding and non-coding transgenes, both in vivo and in vitro. Such systems are useful for the research into toxic or growth inhibitory genetic elements in prokaryote and eukaryote cells. They also provide a convenient way to switch on gene expression at selected timepoints to study the fundamental properties of the induced gene products. However, a critical requirement for inducible gene regulation is lack of expression in the absence of the tetracycline inducer [1]. In addition, the overall responsiveness of the system is critical to avoid toxicity caused by high tetracycline concentrations [1,2].

The reverse tetracycline activator (rtTA) developed by Gossen and Bujard activates gene expression when rtTA fused to the HSV VP16 activator binds to tetracycline response element (TRE) repeats upstream of a minimal promoter [3]. Although inherently less tight than the original Tet-Off system, there have been a number of innovations to improve the tightness of the tetracycline-regulated gene expression [4]. Such approaches have included rtTA sequence alteration to remove nuclease and hairpin structures codon optimisation, amino acid mutations. In the Tet-On promoter region, the sequence and context of tetracycline response element (TRE) repeats, as well as iterations of core promoter elements, have been shown to modulate responsiveness [5,6].

We recently utilised a Sleeping Beauty (SB)-based system for expression of long and complex multi-gene chimeric antigen receptor (CAR) based cassette [7]. This system contained multiple genetic elements, including a reporter, an anti-Her2 CAR, together with an additional gene of interest (GOI: e.g. GFP or Mcl1) as well as the rtTA. We noted significant basal expression and loss of tetracycline-mediated fold-induction in this multigene system. However, more marked improvements were noted by removal of cryptic splice sites with the most profound improvement introduced by a G72V mutation in rtTA-M2 [7,8]. Our earlier study confirmed the suitability of the G72V rtTA mutation, first described in yeast gene-regulation studies [8], for use in mammalian systems. We also confirmed the suitability of minimal promoter elements for tight expression in a Sleeping Beauty transposon cassette but obtained additional improvements through alterations in gene orientation and removal of cryptic splice sites in the rtTA [7].

In this study, we revisited the potential of cryptic splice removal to improve the performance of the rtTA. In our original study, mutation of each and all of the eight cryptic splice acceptor and donor sites resulted in a gain of fold-induction in a complex multi-gene element [7]. In this study, we revisited this approach, but utilised only silent mutations (plus the G72V mutation itself) to abolish the eight cryptic splice sites.

The rtTA sensitivity enhancing (SE) mutations (V9I, F67S, F86Y, and R171K) were identified using an HIV-based in vitro evolution strategy [9]. In this study, combining the SE mutations with G72V rtTA further improved rtTA function through signal to noise boosting, without compromising the low basal expression observed with G72V constructs.

## 2. Materials and Methods

### 2.1. rtTA-M2 Sequence Alterations

Screening for rtTA cryptic splice sites was carried out using the Alternative Splice Site Predictor (ASSP) software and Human Splicing Finder (HSF). Cryptic acceptor and donor splice sites within the original rtTA were predicted based on a criteria of acceptor site cut-off of 2.2 and a donor site cut-off of 4.5 (Appendix A) [10,11]. Splice sites were removed by the introduction of silent mutations within codons that harbour cryptic splice sites or of flanking nucleotides of splice sites (ΔSpl mutations are summarised in Appendix A). Altered sequences were then reanalysed with ASSP software to confirm splice sites removal. The only non-synonymous mutation introduced was the G72V mutation at splice site 215. Sensitivity enhancing (SE) mutations previously developed using an HIV-based in vitro evolution strategy (V9I, F67S, F86Y, and R171K) were introduced into rtTA constructs either alone, or in conjunction with G72V, or both G72V and ΔSpl mutations (Appendix A; Figure 2B) [9].

### 2.2. Vectors

The tetracycline-inducible SB vector (pSBtet-GP) containing the rtTA-M2, reporter gene luciferase and SB transposase containing vector were developed by Eric Kowarz and colleagues and purchased from Addgene [12,13]. SB vectors were modified with alternative rtTA sequences (G72V; SE; ΔSpl; G72V + SE; G72V + ΔSpl; or G72V + SE + ΔSpl; See Figure 1B and Appendix A) by XcmI and XmaI restriction site cloning (gene fragments synthesised by Twist Bioscience, South San Francisco, CA, USA).

### 2.3. Cell Lines and Transfection

The human embryonic kidney 293T (HEK293T) cell line was cultured in Dulbecco’s modified essential medium (DMEM) supplemented with tetracycline-free 10% foetal calf serum (FCS; Pan Biotech, Aidenbach, Germany) and Pen-Strep (100 U/mL penicillin and 100 μg/mL streptomycin; Gibco) at 37 °C with 5% CO_2_. One day prior to transfection, HEK293T cells were seeded in a 6-well plate at 1.5 × 10^5^/mL in 10% FCS/DMEM (without Pen-Strep). HEK293T cells were transfected with a total of 2500 ng of a 4:1 ratio of pSBtet-GP:Luciferase vector to pCMV(CAT)T7-SB100 transposase plasmid using Lipofectamine 3000 (Thermofisher Scientific, Waltham, MA, USA) as per the manufacturer’s protocol. The medium was replaced 18 h post-transfection, cells were then maintained with 2 μg/mL puromycin from 48 h post-transfection for the selection of stably transfected cells until two weeks post-transfection.

Untransfected or stably transfected HEK293T cells (50,000 cells/well in a 96-well plate; 1 × 10^6^/mL) were incubated with a titration of doxycycline (0, 0.1, 0.5, 1, and 5 μg/mL) in 10% FCS/DMEM for 24 h at 37 °C with 5% CO_2_. Luciferase production was analysed by the addition of 100 μL/well of Pierce Firefly Luc One-step Glow assay kit (Thermofisher #16197). Cells were incubated for 1 h at room temperature prior to bioluminescence reading with a Varioskan Lux multimode microplate reader (ThermoFisher, USA). Luciferase data were presented as both relative luminescence units (RLU) and fold change, where fold change was calculated by the RLU of doxycycline treated cells divided by the RLU of untreated cells. Data from three pooled independent experiments were presented as means with error bars representing the standard deviation using GraphPad Prism v9.4.0 (GraphPad Software, San Diego, CA, USA).

In a pilot experiment designed to determine if transposase-dependent SB integration had occurred, HEK293T cells were transfected as described above with 2500 ng total DNA of pSBtet-GP:Luciferase vector with or without the pCMV(CAT)T7-SB100 transposase plasmid. Cells were analysed by flow cytometry (BD FACS Canto II) for GFP expression at days 2, 6, 9 and 14 post-transfection. Cells were then treated on day 14 with 2 μg/mL puromycin for seven days. Following this, cell viability was determined via a haemocytometer with trypan blue staining prior to flow cytometric analysis for GFP expression. Data were analysed with FlowJo v10.7.2 (FlowJo, BD—Advancing the world of health, East Rutherford, NJ, USA).

## 3. Results


*Design and testing of the rtTA-M2 variants:*


### 3.1. Stable Integration of Sleeping Beauty Transposon Containing the Inducible Tet-On Cassette

The primary aim of this study was to optimise the Tet-On system for stably integrated genetic elements. Because inducible gene regulation might vary between transiently transfected vector and stably integrated Sleeping Beauty transposon units, we first analysed the kinetics of Sleeping Beauty transfer vector loss as well as the stable integration of the ITR-flanked transposon unit into the genome of HEK293T cells. To distinguish between transient and integrated gene transfer, transfections were carried out in the presence or absence of a vector encoding the SBX100 Sleeping Beauty transposase. We utilised the transfer vector pSB-Tet that contains the entire inducible Tet-On system, in addition to the RPBSA promoter driving both GFP and puromycin expression (Appendix A). Following transfection in the absence of puromycin selection, GFP expression reduced considerably in the first 7 days, even in the cells co-transfected with the transposase (Figure 1A, Appendix A). However, when puromycin was added to select for stably integrated cells, only cells from the transposase + pSB-Tet cultures remained viable and expanded with detectable GFP expression (Figure 1B,C). Cells transfected with the transfer vector alone did not recover following addition of puromycin (Figure 1B). This was an analytical set up only, and all the cell lines used for the experiments shown in Figure 2 and Figure 3 were generated by selection in puromycin 24 h after co-transfection and maintained for 14 days.

### 3.2. Optimisation of rtTA-M2 Variants in a Stably Integrated Sleeping Beauty Transposon Containing an Inducible Tet-On Cassette

We aimed to improve the current rtTA activity through the comparison of multiple combinations (or alone) of the G72V mutation, SE mutations, and/or removal of additional cryptic splice sites (ΔSpl) from the rtTA-M2 expressed in stably transposed cells. The strategy for the introduction of splice site mutations is shown in Appendix A.

The construct design is summarised in Figure 2. Two constructs, G72V + SE rtTA and G72V + SE + ΔSpl rtTA, were superior to the WT rtTA with average fold change inductions of 7300 and 7100, respectively, compared to WT rtTA, with an average fold change of 3900 (in response to 0.5–5 μg/mL doxycycline). The ΔSpl construct showed only slightly less activity compared to WT rtTA. In contrast, G72V rtTA showed an obvious loss in sensitivity with a substantial drop in performance at doxycycline concentrations less than 5 μg/mL, compared to the WT rtTA. No difference between the G72V and G72V + ΔSpl constructs was noted (Figure 3).

Although the introduction of SE mutations alone into the rtTA-M2 construct resulted in greater levels of luciferase induction at lower levels of doxycycline, the basal expression compromised the fold-induction in transductants housing only the rtTA-M2 SE mutations (Figure 3 and Figure 4). The addition of G72V to the SE-mutant rtTA mitigated the undesirable basal expression of the SE rtTA-M2 noted in the absence of doxycycline (Figure 4). Ablation of cryptic splice sites did not appear to have an impact on absolute RLU or fold-expression values. Nevertheless, the potential of cryptic splice site removal for optimisation of this Tet-On system in other constructs is outlined in Section 4.

## 4. Discussion

Our study confirmed the usefulness of the G72V mutation [7,8] in combination with SE mutations [9] to improve Tet-On performance in a transposon gene-transfer system in a mammalian cell line. Previous studies have demonstrated an improvement of fold-induction of Tet-On systems using the tight M2-TA system with either G72V or G72P mutations in the rtTA gene in yeast and mammalian cells [7,8]. Despite a dramatic lowering of the background signal in the absence of an inducer, both previous studies noted that the rtTA G72 mutation caused a loss in dynamic expression towards higher levels of tetracycline inducer. In a yeast system, Roney et al. showed that the introduction of sensitivity enhancing mutations [9] mitigated the loss in signal resulting from the tightness-enhancing G72V/G72P mutations.

In contrast to Roney et al., who used fluorescent proteins used as reporter genes in yeast, we utilised luciferase in a mammalian transposon system. Interestingly, SE mutations alone improved sensitivity and absolute gene expression (as determined by absolute relative light units), but resulted in undesirable basal expression, thereby reducing the fold expression values. Because our current study compared rtTA solely modified by SE mutations to the other constructs, we were able to determine that the leaky reporter gene expression caused by SE mutations was rescued by the further addition of the G72V mutation. A combination of SE and G72V mutations lead to a substantial improvement in lowering basal level expression, while raising overall gene expression levels.

In our earlier study, it was not clear whether the improvement of G72V was due to the replacement of glycine with a more hydrophobic residue, or alternatively the removal of a cryptic splice site at position 215 nt or a combination of both outcomes [7]. Indeed, removal of additional splice sites improved the performance of the Tet-On system for driving expression of a multi-gene cassette (GFP-P2A-CAR-P2A-rtTA-M2). In our previous study, a E71D mutation was used to test the hypothesis that the loss of the 215 bp cryptic splice site was responsible for the positive effect on rtTA activity by the introduction of the G72V mutation. However, E71 and G72 amino acids create the turn between α-helix-4 and 5 that appears to be a critical determinant of rtTA drug-responsiveness. In this study, we therefore reverted the E71D mutation to its original codon (encoding E71) in order to test the effect of additive effects of other cryptic splice sites mutations with the G72V mutation [7]. Using only silent mutations to ablate potential acceptor and donor splice sites, we did not observe an alteration in inducibility in this simple transposon system. Silent removal of seven potential acceptor and donor splice sites did not alter inducibility or overall dynamics of the luciferase expression, even in the presence of the G72V mutation. Nevertheless, we still recommend using the cryptic splice site-ablated sequences detailed here for driving the expression of other genetic constructs, especially poly-cistronic or otherwise complex mRNA, such as those used in our earlier study [7]. It is possible that cryptic splice sites in rtTA could couple with adjacent acceptor and donor splice sites within the Sleeping Beauty cassette.

The Sleeping Beauty transfer vector and optimised SB100X transposase utilised here likely results in multiple insertions within the genome. Although in difficult-to-transpose cell types, e.g., primary human T cells, integration is likely to be around one site per genome [14], in cells highly permissive to DNA uptake, such as HEK293T, 6–10 insertions per genome have been estimated, with use of higher levels of transposase resulting in toxicity [13,15]. Therefore, in this study, gene expression from the Tet-On cassette likely reflects activity from a number of integrated sites in the genome. The potential relationship between rtTA integration number/expression levels and regulation is intriguing, and rtTA expression levels alone do not seem to predict the inducibility of a Tet-On system. For example, in an auto-regulatory system where rtTA expression driven by the TCE-promoter and where expression was likely to be minimal in the absence of an inducer, we and others have noted the tightest control of gene expression [7,16]. Since the organisation of independent genetic elements within a SB CAR cassette altered the performance of the Tet-On system, we reasoned that the mRNA structure is also a critical consideration for the optimal performance of the Tet-On system. Optimised settings for other applications may require some reorganisation of the genetic structure of the Tet-On elements and reporter/inducible genes. For example, TCE promoter placement and orientation relative to stronger promoters is a critical consideration to avoid unwanted interference. [7,13,17]. The gene transfer system of choice in this study was SB, as we have previously noted a loss of activity using lentiviral vectors that may always be optimised for inducible expression (data not shown). For example, viral LTR (even in SIN vectors) still possess detectable transcriptional activity [18] that might modulate inducible activity from the Tet-On system. To maximise inducibility, the SB-based Tet-On system developed by Kowarz et al. contains flanking insulators to prevent read-through interference from adjacent genomic sites and transcription factor binding sites within SB were removed by selective mutation. In addition, rtTA expression is driven by the RPBSA synthetic promoter that lacks enhancer activity, thereby preserving the tightness of the TCE promoter [13].

In this setting, cryptic splice site ablation from rtTA-M2 did not have an effect on gene regulation, but given our previous findings, we recommend the use of G72V/SE variant rtTA-M2 with cryptic splice site removal. Synonymous codon optimisation can have deleterious effects on gene expression due to alterations in translation speed impacting protein folding, glycosylation and nascent protein localisation [19,20]. Bearing in mind that the rtTA-M2 gene has already been codon optimised for removal of endonuclease sites potential hairpin, identified splice sites and translation in human cells, caution should be applied to further codon adaptation with index-based tools since this can lead to the introduction of additional cryptic splice sites and/or complete loss of rtTA-M2 activity, as observed in our earlier study [7]. In conclusion, the results emphasise the superiority of the G72V mutation [8] in combination with SE mutations [9] for improvement of the rtTA system. The G72V/SE rtTA-M2 variant offers advantages over the original rtTA-M2 for driving inducible gene expression and will be useful for the responsive control of mammalian gene expression, including the tight control of toxic coding and non-coding sequences.

## Figures and Tables

**Figure 1 genes-13-01679-f001:**
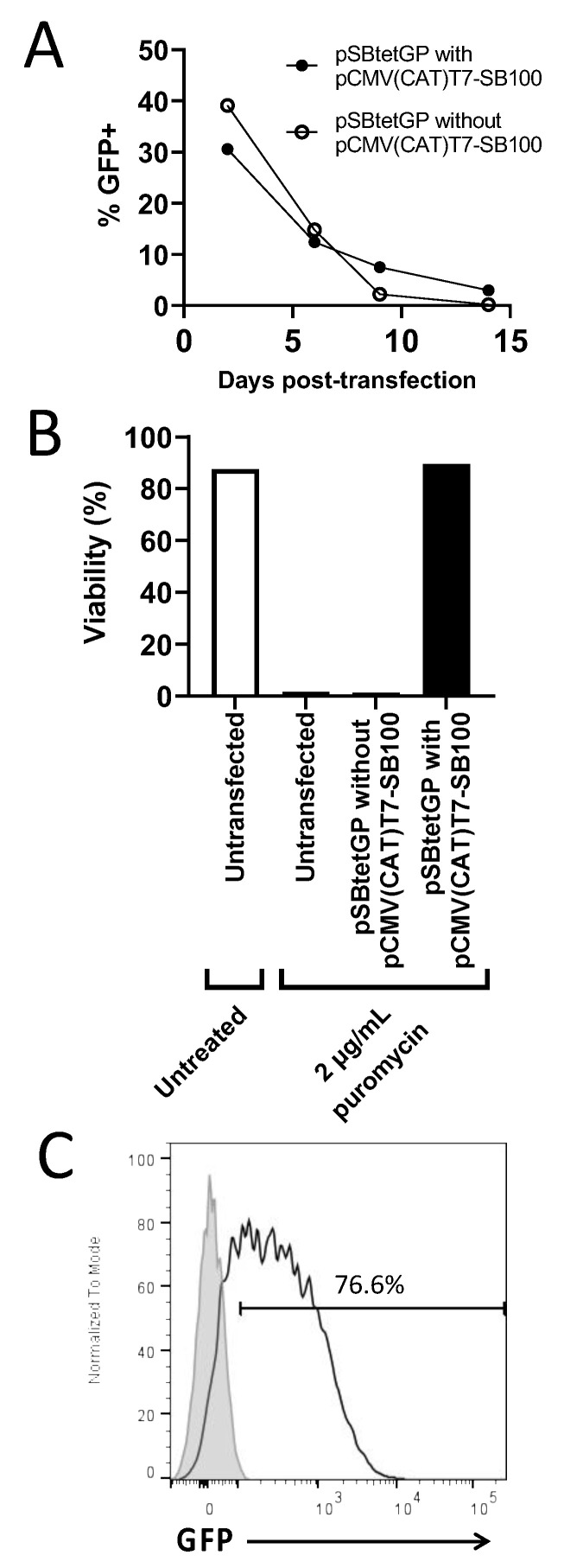
**Assessing integration of the SB vector:** (**A**) HEK293T cells were lipofectamine 3000 transfected with pSBtetGP:Luciferase plasmid with or without the pCMV(CAT)T7-SB100 transposase containing plasmid. HEK293T cells were analysed on days 2, 6, 9 and 14 for GFP expression by flow cytometry. (**B**,**C**) Untransfected or transfected HEK293T cells at day 14 were then treated with or without 2 µg/mL puromycin as stated for seven days. (**B**) Cell viability was then analysed by trypan blue staining, and (**C**) HEK293T cells transfected with pSBtetGP:Luciferase and the pCMV(CAT)T7-SB100 plasmids (black line) or untransfected control (grey shaded peak) were analysed for GFP expression by flow cytometry.

**Figure 2 genes-13-01679-f002:**
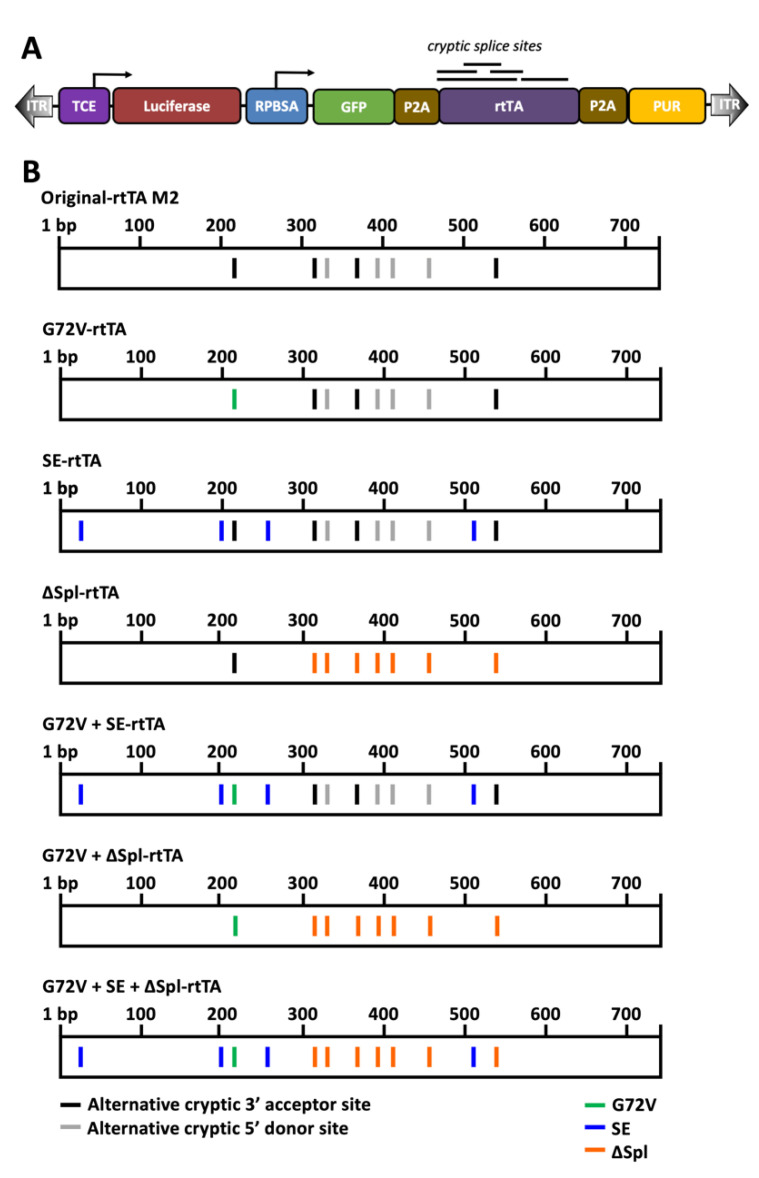
**Overview of rtTA constructs investigated in this study.** (**A**) Schematic of construct design derived from the original pSBtet-GP developed by Kowarz et al. ITR: inverted terminal repeat; TCE: tetracycline responsive promoter; GOI: gene of interest (firefly luciferase); PA: polyadenylation site; rtTA: reverse tetracycline-controlled transactivator. (**B**) Summary of rtTA constructs utilised within this study: Original-rtTA M2 (wildtype); G72V-rtTA; SE-rtTA; ΔSpl-rtTA; G72V + SE-rtTA; G72V + ΔSpl-rtTA; and G72V + SE + ΔSpl-rtTA. Black indicates predicted alternative cryptic 3′ acceptor sites; grey indicates predicted alternative cryptic 5′ donor sites; green indicates G72V mutation; blue indicates SE mutations; and orange indicates ΔSpl mutations.

**Figure 3 genes-13-01679-f003:**
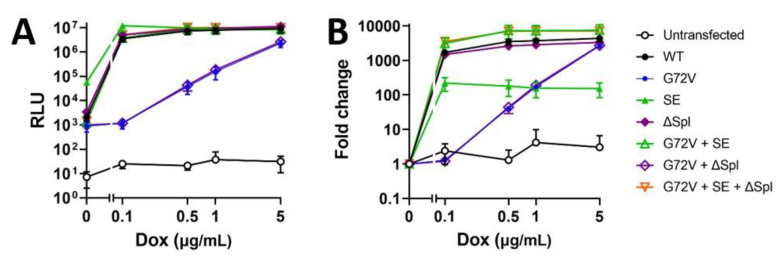
**Enhanced sensitivity mutations with G72V optimises rtTA-M2 inducibility**. HEK293T cells were stably transposed with pSBtet-GP:Luciferase with wildtype (WT) rtTA, or rtTA containing the G72V mutation, high-sensitivity (SE) mutations, cryptic splice sites removed (ΔSpl), G72V and SE, G72V and ΔSpl, or G72V with both SE and ΔSpl. Untransfected or transfected HEK293T cells were incubated with a titration of doxycycline (Dox; 0, 0.1, 0.5, 1, or 5 μg/mL) for 24 h prior to the analysis of luciferase production. (**A**) Relative luminescence units (RLU) or (**B**) the luciferase fold change compared to nil doxycycline addition is shown with the mean and SD of three pooled experiments.

**Figure 4 genes-13-01679-f004:**
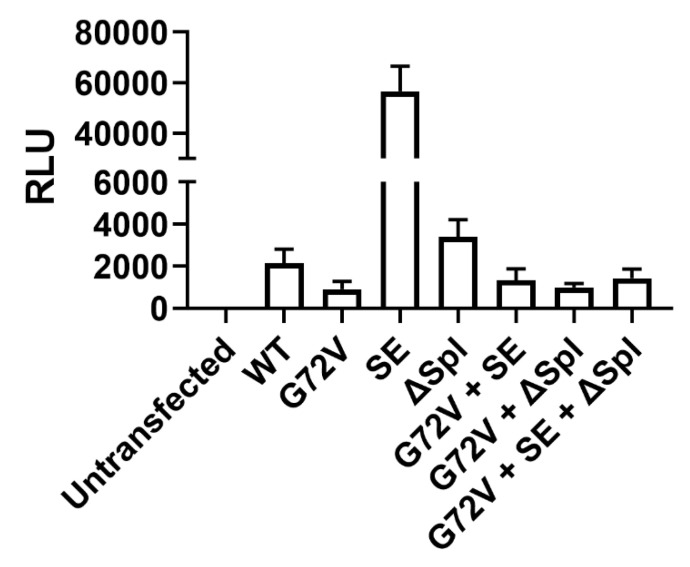
**G72V minimises basal expression even in the presence of high sensitivity mutations within the rtTA**. HEK293T cells untransfected or stably transposed with pSBtet-GP:Luciferase with WT, G72V, SE, ΔSpl, G72V and SE, G72V and ΔSpl, or G72V with both SE and ΔSpl were analysed for basal levels of luciferase production. Relative luciferase units (RLU) are shown with the mean and SD of three pooled experiments.

## Data Availability

Additional data or reagents are available from the authors on request. The optimised rtTA-M2 (SE, G72V, ΔSpl) and Vectors will be available from Addgene. All sequences of the rtTA-M2 variants are given in the Appendix A.

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
