# Peer review of "Noise-Reduction and Sensitivity-Enhancement of a Sleeping Beauty-Based Tet-On System"

_genes, 2022, doi:10.3390/genes13101679_

Round 1

Reviewer 1 Report

This paper describes further improvement in the tet-system commonly used to control gene expression. The work is done in mammalian cells. It seems that the improvement is a real phenomenon and can be useful generally for researchers needing a tight control and good responsiveness for their gene expression of interest.

Some comments:

Lines 35-40. The list is difficult to understand as it is written. Make the message more explicit. 

Supplementary fig 2 contains extra material in its legend (from Fig 1), and needs to be cut off.

Table 1 would be better suited in "Supplementary files".

The data is based on the assumption that the constructs used are integrated. With different constructs, the integration sites do vary. This means that with a low frequency of integration, the actual integration sites may matter. Are there any indications (e.g. from previous literature) of how many integrations per transfection is expected? If it is "a lot", many different sites even up the result, which is then OK. At least this should be discussed.

To absolutely guarantee that the results are reliable with this respect, simply repeating the experiment would be pertinent. However, this is not needed, if the abovementined considerations indicate that integrations occur at a high rate.             

Author Response

Thank you for these comments.

Lines 35-40. The list is difficult to understand as it is written. Make the message more explicit. 

We have altered the descriptions of improvements to the Tet-On system by rewording and splitting this into two sentences. An additional reference has been added to properly describe alterations to the promotor region: "Such approaches have included rtTA sequence alteration to remove nuclease and hairpin structures codon optimisation, amino acid mutations. In the Tet-On promoter region, the sequence and context of tetracycline response element (TRE) repeats, as well as iterations of core promoter elements have been shown to modulate responsiveness [5,6]."

Supplementary fig 2 contains extra material in its legend (from Fig 1), and needs to be cut off.

Corrected.

Table 1 would be better suited in "Supplementary files".

Table 1 has been moved to Supplementary material (now Suppl. Table 1)

The data is based on the assumption that the constructs used are integrated. With different constructs, the integration sites do vary. This means that with a low frequency of integration, the actual integration sites may matter. Are there any indications (e.g. from previous literature) of how many integrations per transfection is expected? If it is "a lot", many different sites even up the result, which is then OK. At least this should be discussed.

We presented data in support of our proposal the that transposon units are indeed integrated (Suppl. Figure 2 and Suppl. Table 2). This assumption is based mainly on the low level gene transfer of GFP and puromycin occurring in transfections carried out in the absence of the SB100X transposase gene. Selection with antibiotics likely increases the chances of isolating cell lines with a high number of integrations. However, integration sites were not identified or quantified in this study.  We now further discuss SB integration sites in the Discussion:

"The Sleeping Beauty transfer vector and optimised SB100X transposase utilised here likely results in multiple insertions within the genome. Although in difficult-to-transpose cell-types, e.g. primary human T cells, integration is likely to be around one site per genome [14], in HEK293T cells which are highly permissive to DNA uptake, 6-10 insertions per genome have been estimated, with higher levels of transposase resulting in toxicity [13,15]. Therefore, in this study, gene expression from the Tet-On cassette likely reflects activity from a number of integrated sites in the genome. The potential relationship between rtTA intregration number / expression levels and regulation is intriguing, and rtTA expression levels alone do not seem to predict inducibility of a Tet-On system. For example, in auto-regulatory systems where rtTA expression driven by the TCE-promoter was likely to result in be minimal rtTA expression in the absence of inducer, we and others have noted the tightest control of gene expression [7,16]."

To absolutely guarantee that the results are reliable with this respect, simply repeating the experiment would be pertinent. However, this is not needed, if the above mentioned considerations indicate that integrations occur at a high rate.         

Noted. 

Reviewer 2 Report

In the manuscript I received for the revision, Saunderson and colleagues intended to improve the signal-to-noise ratio and the sensitivity of the Tet-On system. Drug-induced expression systems allow for robust and stringent gene regulation in functional genetic studies and gene therapy applications.

In this work, the Authors were focused on the optimisation of the rtTA component of the Tet-On system, in which they introduced G72V mutation, sensitivity enhancing mutations and ablated cryptic splice sites. Although the last approach does not seem to affect gene regulation, other results presented in the manuscript indicate the advantage of combined G72V and sensitivity-enhancing mutations over the original Tet-On system.

The manuscript is well written, and the conclusions are clearly derived from experimental results. I would only reconsider the title of this manuscript and remove the reference to the Sleeping Beauty transposon system, as the findings presented in this work can also be relevant for both viral and non-viral systems, where DOX-inducible regulation is used.

Author Response

The manuscript is well written, and the conclusions are clearly derived from experimental results. I would only reconsider the title of this manuscript and remove the reference to the Sleeping Beauty transposon system, as the findings presented in this work can also be relevant for both viral and non-viral systems, where DOX-inducible regulation is used.

Thank you for these comments. We would prefer to emphasise the use of this Sleeping Beauty Tet-On system (and retain mention of this in the title) for the following reasons now clarified in the discussion:

"The gene transfer system of choice in this study was SB, as we have previously noted a loss of activity using lentiviral vectors which may always be optimised for inducible ex-pression. For example, viral LTR (even in SIN vectors) still possess detectable transcriptional activity [18] that might modulate inducible activity from the Tet-On system. To maximise inducibility, the SB-based Tet-On system developed by Kowarz et al. contains flanking insulators to prevent read-through interference from adjacent genomic sites. In addition, in rtTA expression is driven by the RPBSA synthetic promoter that lacks enhancer activity thereby preserving the tightness of the TCE promoter [13]."